# Endometriosis-Associated Ovarian Cancer: What Are the Implications for Women with Intact Endometrioma Planning for a Future Pregnancy? A Reproductive Clinical Outlook

**DOI:** 10.3390/biom12111721

**Published:** 2022-11-21

**Authors:** Johnny S. Younis

**Affiliations:** 1Reproductive Medicine, Department of Obstetrics and Gynecology, Baruch-Padeh Medical Center, Poriya 15208, Israel; jyounis@poria.health.gov.il; Tel.: +972-505286981; Fax: +972-46737478; 2Azrieli Faculty of Medicine in Galilee, Bar-Ilan University, Safed 1311502, Israel

**Keywords:** endometriosis, endometrioma, endometriosis-associated ovarian cancer, clear-cell ovarian carcinoma, endometrioid ovarian carcinoma, genetic profiles, reproductive age, ovarian reserve, ovarian reserve biomarkers, transvaginal ultrasound, magnetic resonance imaging

## Abstract

Endometriosis is a chronic, universal, and prevalent disease estimated to affect up to 1:10 women of reproductive age. Endometriosis-associated ovarian cancer (EAOC) developing at reproductive age is challenging and of concern for women and practitioners alike. This outlook review focuses on the occurrence of EAOC, especially in infertile women or those planning for a future pregnancy, from the perspective of a reproductive endocrinologist, based on recent evidence. Contemporary pathogenesis, genetic profiles, evidence of causality, clinical diagnosis, prognosis, and up-to-date management are discussed. EAOC seems to be merely associated with endometrioma and includes clear-cell and endometrioid ovarian carcinoma. Although endometrioma is frequently found in women of reproductive age (up to 1:18 of women), EAOC appears to be a rare occurrence. These women are of more advanced reproductive age, nulliparous, and hyperestrogenic, with a large-sized unilateral endometrioma (>9 cm) containing solid components and papillary projections. Each case suspected to have EAOC has specific characteristics, and a multidisciplinary discussion and appropriate patient counseling should be conducted to reach an optimal therapeutic plan. Since most of these cases are diagnosed at an early stage with a favorable prognosis, fertility-sparing surgery may be feasible. The pros and cons of fertility preservation techniques should be discussed.

## 1. Introduction

Endometriosis is a chronic, inflammatory, universal, and widespread disease that seems ubiquitous in women of reproductive age, with an estimated prevalence of one in 10 women [1]. Alternately, it may affect as many as 20–30% of women with subfertility and 40–60% with chronic pelvic pain [2,3]. Endometriosis primarily affects the female pelvis external to the uterus and may extend to the peritoneal cavity and beyond. Pelvic pain and subfertility are the most prominent manifestations of endometriosis, with an adverse impact on quality of life, impending need for assisted reproduction technology (ART) treatment, and economic burden on the health system. Nonetheless, almost 50% of affected women may be asymptomatic and oblivious to the disease. There are three established subtypes of the disease—superficial, endometriotic ovarian cyst, and deep infiltrating—though endometrioma is the most pathognomonic and diagnosed form. According to the revised classification of endometriosis by the American Society of Reproductive Medicine, endometrioma represents a more advanced stage of the disease [4]. Endometrioma affects up to 55% of women unilaterally and 28% bilaterally [5,6], an equivalent estimation of 1:18 and 1:36, respectively, in women of reproductive age.

It has long been known that endometriosis has several pathogenic features of cancerous development, containing chronic inflammatory environment, invasion of other tissues, growth into local and distant loci, high recurrence rate, active angiogenesis, and resistance to apoptosis [7,8]. Nonetheless, until recently, endometriosis was considered a benign estrogen-dependent disease that artlessly resolves at the end of reproductive age. In the last few years, new data have been accumulating to corroborate endometriosis as associated with an increased ovarian malignancy risk, termed today as endometriosis-associated ovarian cancer (EAOC), which may even extend beyond perimenopause (discussed in Section 4).

EAOC developing in reproductive age may raise much concern, especially for infertile couples and women postponing live birth or planning for a future pregnancy. Furthermore, for clinicians, these concerns promote practical issues for the long-term management of women with ovarian endometriosis through adulthood and past the menopausal transition. However, this link and its translation into clinical practice in terms of information to patients and early cancer detection still need to be clarified [9]. This outlook review explores the occurrence of EAOC in reproductive age, contemporary pathogenesis, genetic profiles, evidence of causality, current clinical diagnosis, prognosis, and implications for women with an intact endometrioma planning for a future pregnancy.

## 2. Materials and Methods

To achieve the objectives for this study, I first conducted a broad search of the published English literature on Pubmed.com (accessed on 2 July 2022) from January 2015 to June 2022 with the keywords ‘endometriosis’, ‘endometrioma’, ‘ovarian endometriosis’, ‘atypical endometriosis’, ‘ovarian cancer’, ‘endometriosis-associated ovarian cancer’, ‘clear-cell ovarian carcinoma’, ‘endometrioid ovarian carcinoma’, ‘imaging’, ‘ultrasonography’, ‘magnetic resonance imaging’, ‘biomarkers’, ‘reproductive age’, ‘ovarian reserve’, ‘infertility’, ‘pregnancy’, ‘live birth’, ‘systematic review’, and ‘meta-analysis’. The relevance of the publications was evaluated upon reading the abstract. A manual search of review articles and cross-references completed the investigation. Articles with an inappropriate design were excluded.

## 3. What Is the Risk of EAOC?

Previous publications may have confused patients and practitioners concerning the risk of cancer in women with endometriosis, mainly due to the intricacy of this topic and the controversy in the literature. One large previous cohort study, including 37,434 women, did not find an increased risk of ovarian carcinoma or other malignancies among women with endometriosis [10]. Furthermore, systematic reviews and meta-analyses published in the last decade evaluating cancer risk in women with endometriosis, resulted in contrasting estimates with high heterogeneity [11,12,13,14,15]. Potential confounders that may have affected risk estimation in EAOC are summarized in Table 1 [16,17,18,19,20].

In a recent, carefully conducted systematic review and meta-analysis exploring the association between endometriosis and cancer risk, 49 population-based case-control and cohort studies were included [16]. The results were analyzed to account for the impact of methodological confounders and risk of bias among eligible studies and to explore the disease heterogeneity in different cancer types and endometriosis subtypes. Random effects meta-analysis was used to estimate summary relative risks (SRR) and confidence intervals (CI). Devoting this methodological attention, about half of the included publications in the quantitative analysis had a severe or critical risk of bias.

Cancer-specific analyses showed a positive association between endometriosis and ovarian cancer risk (SRR = 1.93, 95% CI = 1.68–2.22). This association for EAOC was the strongest for clear-cell (SRR = 3.44, 95% CI = 2.82–4.42) and endometrioid (SRR = 2.33, 95% CI = 1.82–2.98) histotypes. However, the heterogeneity of these estimates was high. An association was also found between endometriosis and low-grade serous tumors but not high-grade serous carcinoma.

Notably, in the meta-analysis by Kvaskoff et al. [16], only four studies provided estimates of the association between endometriosis and ovarian cancer risk by endometriosis subtype [17,22,23,24]. Three of them focused on ovarian endometrioma only, irrespective of other subtypes [22,23,24]. Based on these studies, the SRR for the association between endometrioma and ovarian cancer was 5.41 (95% CI = 2.25–13.0). While explored only by one study, the other subtypes, superficial and deep infiltrating endometriosis, were not linked to EAOC [17]. Further prospective targeted and more powered studies ought to investigate EAOC and its association with the three sub-types of the disease.

To put the risk of EAOC development in perspective, it seems more edifying to calculate the women’s lifetime ovarian cancer risk. According to 2017–2019 data from the National Cancer Institute, lifetime ovarian cancer risk in the general female population is 1.1% [25]. Considering the SRR of 1.93 in women with endometriosis, the calculated lifetime ovarian cancer risk in women with endometriosis is estimated at 2.1%, which seems low. To place the threat of EAOC in proportion with other malignancy risks in women, the lifetime risk of cancer for breast is 12%, lung at 6%, and bowel at 4%, which is much higher than the risk of developing ovarian cancer.

## 4. When Is EAOC Diagnosed?

Most cases of ovarian cancer in the general population are diagnosed in women past reproductive age, largely postmenopausal, yet 12.1% of ovarian cancer patients are <44 years of age [26]. Many of these young women will present with borderline tumors. Furthermore, they include cases of non-epithelial ovarian cancer, comprising almost 10% of all cases of ovarian cancers, and typically present at an early age [27].

The likelihood of the epithelial EAOC during the reproductive period seems infrequent. The literature contains several case reports of sporadic occurrences [28,29,30,31]. In a recent single-center retrospective cohort study conducted in one of the largest endometriosis medical centers in China that targeted women below 40 years of age, only 40 cases of EAOC were found over 12 years [32]. This cohort of women with EAOC represented 42.6% of the total number of women with ovarian carcinoma at a young age in the exact center. The mean age of these women was 34.4 ± 4.0, ranging from 27 to 40 years. About half of the women in this study did not have children upon diagnosis.

Another retrospective Chinese multi-center cohort study targeted 237 women with stage I ovarian clear-cell carcinoma, summarizing 20 years of experience. The mean patient age at initial diagnosis was 48.9 ± 11.0 years, and 146 patients (61.6%) were premenopausal [33]. Among this cohort of women, 105 (44.3%) had histological features compatible with EAOC. Compared to controls, women with EAOC were diagnosed more frequently in the premenopausal years and at the early stages of the disease.

Conversely, in a recent large population-based Dutch nationwide cohort study with a validated database of 131,450 women with histological diagnosis of endometriosis compared to 132,654 matched controls, the study confirmed the significantly higher incidence of clear-cell and endometrioid ovarian cancer [34]. Although the median age at ovarian cancer diagnosis was earlier in the endometriosis women as opposed to controls, corresponding to 56 years (interquartile range, 49–63) and 60 years (interquartile range, 53–67), respectively (*p* < 0.05), this is well beyond the reproductive years. However, the different delays encountered between clinical and histological endometriosis diagnosis in these women may have been caused by the increased age at histological diagnosis in this study.

Notably, in this study, endometriosis and EAOC were diagnosed synchronously in many of these women after the average menopausal age [34]. This may imply that EAOC risk remains, even when clinical endometriosis symptoms have ceased, suggesting the need for long-term follow-up and counseling.

Collectively, in most cases, ovarian epithelial carcinoma is a disease of postmenopausal age. It is estimated that 1:8 of women with ovarian cancer may develop the disease during their reproductive years. However, many will have borderline or non-epithelial tumors. Cases of EAOC during the reproductive years seem to be infrequent. About 50% of EAOC cases present at premenopausal age but not necessarily during the reproductive period. Half of these cases may have no children upon diagnosis.

## 5. What Is the Pathogenesis of EAOC?

### 5.1. Background

Clear-cell and endometrioid carcinomas are the most intensely and reproducibly associated malignancies with endometriosis. Coexistence with endometriosis is observed in 21–51% of patients with clear-cell carcinoma and 23–43% with endometrioid carcinoma [35,36,37]. While endometriosis is also associated with low-grade serous ovarian carcinoma, this linkage needs to be more defined. In the last few years, genuine efforts have been invested in exploring the pathogenesis of EAOC occurrence and whether there is a causal relationship.

Numerous theories of endometriosis occurrence, development, and expansion are still cited and discussed. Succinctly, they include retrograde menstruation, coelomic metaplasia, circulatory or lymphatic spread, genetic predisposition, and self-abnormalities in the humoral and cell-mediated immune systems. Their relation to the development of EAOC is still under investigation.

Genetic, inflammatory (including free-iron-induced oxidative stress), immunological, and hormonal factors have been implicated in the malignant transformation of endometriosis [38]. However, the pathogenesis of EAOC is still an unresolved enigma and has been a matter of active investigation and debate over the last few years. Several recent reviews, systematic reviews, and meta-analyses targeting this topic did not reach definite conclusions [38,39,40,41].

A recent systematic review has addressed the topic of cancer-associated mutations (CAMs) in endometriosis patients, shedding some light on the pathogenesis and pathophysiology of the disease. However, it must still achieve a clear picture or definitive conclusion [42]. Furthermore, CAMs do not necessarily lead to malignant transformation, as it requires gaining and accumulating the precise type in a specific combination of CAMs to complete the malignant transformation.

An innovative dual paradigm has been developed to gain insight into and explore the composite molecular genetic pathways implicated in the pathogenesis of primary ovarian epithelial carcinomas and to elate these pathways with histopathological classification [43,44]. This dual model suggests two distinct types of ovarian epithelial carcinoma with different molecular profiles, type I and type II. Type I presents at an early stage with low-grade features, including clear-cell, endometrioid, and low-grade serous carcinomas. Type I frequently arises through a defined sequence, either from endometriosis or borderline serous tumors. Type II carcinoma is a much more frequent disease and usually presents at advanced stages. Type II are typically high-grade serous carcinomas, arising in most cases from the fimbriated end of the fallopian tubes, as foci of small in-situ tubal intraepithelial carcinoma [45], with silent progression, peritoneal seeding, and fast spread.

Indeed, the molecular profiles of these two types seem to be different and correlate well with the distinct nature of type I and type II carcinomas. Recent molecular studies in type I ovarian carcinomas identified somatic mutations in *ARID1A*, *KRAS*, *PTEN*, *PIK3CA*, *MLH1*, and B catenin [46,47]. In addition, *TP53*, *BCL2*, and *POLE* mutations have also been described [48,49].

In contrast, most type II tumors are characterized mainly by *TP53* mutations. In fact, according to the Cancer Genome Atlas dataset, the *TP53* mutations are present in almost 96% of high-grade serous ovarian carcinomas [50]. Nevertheless, *TP53* mutation, otherwise pathognomonic for high-grade serous ovarian carcinoma, is found in 30% of endometriosis associated with clear cell carcinoma. Benign endometriosis has not been associated with *TP53* mutation, nor has it been found in endometriosis coexisting with endometrioid carcinoma [51].

To further explore the pathogenesis of malignant transformation of endometriosis to EAOC, a recent study evaluated the genomic-wide functions involved with data-driven analysis based on the functionomes of endometriosis, clear-cell ovarian carcinoma, and endometrioid ovarian carcinoma [52]. This was achieved by studying the microarray gene expression datasets of these three illnesses, from the National Center for Biotechnology Information Gene Expression Omnibus, with the quantified molecular functions defined by 1454 Gene Ontology term gene sets. This study demonstrated that deregulated metabolism, cell cycle control, cell–cell signaling, hormone activity, inflammatory response, immune response, and oxidoreductase activity are vital components of EAOC pathogenesis.

Furthermore, several studies have suggested that atypical endometriosis, characterized by cytological and architectural atypia, hyperplasia, large nuclei, and increased nuclear–cytoplasmic ratio, may be a direct precursor of EAOC [53]. A recent systematic review of molecular biomarkers of atypical endometriosis, summarizing 39 eligible studies, has found a high heterogeneity among the reports [54]. Nevertheless, certain constancy was detected for altered expression in phosphoinositide *3-kinase (PI3K)/AKT/mTOR* pathway, *ARID1a*, estrogen and progesterone receptors, and transcriptional, nuclear, and growth factors in such cases. The authors concluded that since these biomarkers involve expensive molecular analysis and none has solid evidence, there is no justification for their regular application in the clinical setting.

### 5.2. What Is the Evidence for Causality in EAOC?

Currently, advanced human genetics seems to be the best tool to explore complex and heterogeneous disease causality in modern medicine. Genetics provides a robust scientific platform for establishing a relationship between a cause and an effect—in this case, endometriosis and EAOC—however, causality in genetics is probabilistic and rarely a deterministic certainty. The causal relationship between a genetic variant and a phenotype is provisional and based on the conditions and the environment, such as the genetic backgrounds in which the causal variants and the phenotype operate. These fundamental aspects seem to apply to the composite pathogenesis of EAOC.

Genetic studies published to explore common genetic grounds between endometriosis and ovarian cancer have employed three different strategies. The first inspect common alleles associated with different histotypes of epithelial ovarian cancer, pooling data from multiple genome-wide genotyping projects or utilizing the Mendelian randomization methodology to look for germline genetic variants as proxies for causal effects of risk factors [55,56]. The second assesses the association between endometriosis and ovarian cancer as a distinct disease [57,58], while the third explores the link between endometriosis and specific histotypes of ovarian cancer [59,60]. Each strategy adds value and is complementary to the others. However, since ovarian epithelial cancer is a heterogeneous disease with diverse pathogenesis and pathways, the third strategy seems more appropriate and straightforward for exploring EAOC causality.

One large study explored shared genetic etiologies between two endometriosis databases genotyped on common arrays with full-genome coverage (3194 cases and 7060 controls) and a large ovarian cancer dataset genotyped on the customized Illumina Infinium iSelect (iCOGS) arrays (10,065 cases and 21,663 controls). Evidence was found for shared genetic risks between endometriosis and all ovarian cancer histotypes, except for the mucinous type. Clear-cell carcinoma showed the strongest genetic correlation with endometriosis (0.51, 95% CI = 0.18–0.84) [59].

Just recently, a strong genetic relationship between endometriosis and epithelial ovarian cancers was reported employing state-of-the-art methods, including genetic correlation, Mendelian randomization, bivariate genome-wide association studies, colocalization, and functional genomic analyses [60]. The data explored included 14,949 cases/190,715 controls for endometriosis and 25,509 cases/40,941 controls for ovarian epithelial carcinoma. A significant genetic correlation (rg) was found between endometriosis and clear-cell carcinoma (rg = 0.71) and endometrioid carcinoma (rg = 0.48), verified by Mendelian randomization analysis. Furthermore, a bivariate meta-analysis identified 28 loci associated with endometriosis and ovarian epithelial cancer, including 19 with evidence for a shared underlying association signal.

Collectively, previous epidemiological observations have shown an association between endometriosis and EAOC, specifically with clear-cell and endometrioid carcinomas. Several studies have been conducted to understand the underlying mechanisms of this malignant transformation, suggesting multi-factorial pathways. Employing state-of-the-art genetic methods has provided evidence of genetic correlation and a strong potential causal relationship between endometriosis and EAOC. Future fine-mapping and histotype-specific functional analyses may substantiate EAOC causality. Furthermore, these novel advancements may pave the way for targeted ovarian epithelial cancer screening and facilitate potential preventive pharmacological interventions.

## 6. What Is the Strategy for the Clinical Diagnosis of EAOC in Reproductive Age?

### 6.1. Clinical Manifestations and Risk Factors

The clinical diagnosis of EAOC during reproductive age is challenging since endometrioma is a frequent finding, while EAOC is an unusual complication. While comparable symptoms may result from both entities, relapsing or worsening pelvic pain symptoms should increase the index of suspicion. The rapid growth of endometrioma size may be suspicious as well. Clinical risk factors implicated in the development of EAOC include women with endometrioma above 45 years of age, nulliparity, larger size endometrioma (>9 cm), hyperestrogenism (endogenous or exogenous), or an endometrioma with a solid component [40].

### 6.2. Ovarian Cancer Biomarkers

Ovarian cancer biomarkers seem to have no added value in diagnosing EAOC. In general, serum cancer antigen 125 (CA-125) is elevated in about 80% of epithelial ovarian cancer and is commonly endorsed for monitoring the response to treatment. It is particularly reliable for high-grade serous ovarian cancer, mainly in the postmenopausal period. The test is frequently used off-label in clinical practice to help categorize adnexal masses [61]. The performance of the test improves when it is combined with pelvic ultrasound.

On the other hand, serum CA-125 testing has a very low sensitivity for stage I disease in those with epithelial subtypes of cancer different from high-grade serous adenocarcinoma and in the premenopausal period [62]. Furthermore, serum CA-125 may be elevated in cases with benign endometrioma, though there is no evidence for it as a diagnostic of endometriosis [63]. Thus, CA-125 testing does not seem valuable in diagnosing EAOC, especially at reproductive age.

Human epididymis protein 4 (HE4) is a novel tumor biomarker approved for determining the likelihood that an ovarian mass is cancerous, with a sensitivity similar to that of CA-125 but with superior specificity [64]. In a recent small cohort of 76 women, 59 with endometrioma and 17 with EAOC, serum CA-125, and HE4 levels did not differ between the two groups [65].

### 6.3. Transvaginal Ultrasound

Generally, imaging has a vital role in the clinical evaluation and differential diagnosis of female ovarian or adnexal findings, noticeably in reproductive age. Pelvic transvaginal ultrasound (TVUS) is the first-line imaging examination for a suspected ovarian tumor in the diagnosis of malignancy [66], including differentiating benign endometrioma from EAOC. Computerized tomography (CT) scan has poor performance in assessing ovarian mass. However, magnetic resonance imaging (MRI) can be a valuable adjunct for ovarian findings described as intermediate or atypical by TVUS.

The morphological features of the ovarian finding on TVUS are employed to categorize the risk of malignancy. Among several suggested ultrasonographic classification systems, the two most promising tools are the former International Ovarian Tumor Analysis (IOTA) [67] and the recent Ovarian-Adnexal Reporting and Data System [68], summarized in Table 2. In these two classifications, the more complex a mass is, the higher the likelihood it is malignant. Both TVUS systems seem to have good performance, high sensitivity, and specificity in predicting risk estimation in clinical practice.

Benign endometrioma in the reproductive age typically appears on TVUS as an ovarian mass with ‘ground glass’ echogenicity of the cyst fluid and one to four locules (i.e., uni-cystic or multi-cystic), without solid parts or papillations. As age increases, multilocular cysts and cysts with papillations and other solid components become more common, presumably due to retracted blood clots within the endometrioma [69]. In contrast, the ground glass echogenicity of cyst fluid becomes less common, while maximal lesion diameter does not seem to change with age.

In a retrospective multicenter study employing the IOTA system and summarizing 239 women with a histological diagnosis of endometrioid ovarian cancer, TVUS demonstrated usually large, unilateral, multilocular-solid, or solid tumors with low echogenicity of cyst fluid [70]. In this study, in about 25% of patients, cancer developed from endometriosis (EAOC), and 20% of these women had synchronous endometrial carcinoma. TVUS characteristics differed between those with and others without endometriosis, the former being more often unilateral cysts with papillary projections and no ascites.

In another retrospective multicenter study employing the IOTA system, 152 women with clear-cell ovarian carcinoma were analyzed by TVUS. Most tumors were unilateral, particularly large in diameter. All contained solid components, and about one-fourth were entirely solid [71]. Papillary projections were present in almost 40% of cases, with vascularization in most of these projections. In this study, about 20% of analyzed cases were determined to have developed from endometriosis (EAOC). The ground-class appearance was more common in EAOC cases.

In addition, a recent study was published comparing 63 women with atypical endometrioma features on TVUS with 53 cases of clear-cell ovarian carcinoma [72]. In multivariate analysis, advanced age (>47.5 years), large cysts (>11.55 cm), large solid components (size > 1.37 cm), and loss of ground-glass echogenicity were independent factors suggestive of malignancy.

Jointly, endometrioid and clear-cell ovarian carcinomas seem to have similar TVUS features. Most are large and above 9 cm, with a mean size of 11–13 cm [73], unilateral tumors with solid components, and papillary projections (present in 40–50%), especially in EAOC cases. Since papillations are typical TVUS features of serous and mucinous borderline tumors, they should be accounted for in the differential diagnosis. Furthermore, since both endometrioid and clear-cell ovarian carcinomas have similar clinical presentations, it seems challenging to distinguish between these two entities preoperatively.

### 6.4. Magnetic Resonance Imaging

In clinical practice, about 5–25% of cases will have indeterminate adnexal findings by TVUS [74]. The tumors are usually benign in most cases, including those with endometrioma. MRI supportive performance in these cases may reduce patient anxiety, repeat imaging, unnecessary follow-up, and avoid surgery.

Classical benign endometriomas on MRI typically display features of T2-weighted image shading. This refers to their higher signal intensity on T1-weighted imaging caused by the proteinaceous and viscous contents of the endometrioma, which expectedly exhibits T2 signal intensity; lower than that of simple fluid [74].

The enhanced solid portion of the endometrioma combined with a large cyst suggests malignancy [75]. Furthermore, the disappearance of shading within the endometrioma on T2-weighted images may also mean malignant transformation [76]. These changes have been recently confirmed in a 10-year longitudinal follow-up study of 50 women [77].

To differentiate between EAOC and non-EAOC, a recent retrospective study including 54 women was conducted [78]. Although much overlap was found between the two groups, in a multiple logistic regression analysis, unilocular lesions and hypo-intensity on T2 weighted images in cystic components were found to distinguish between the two entities.

MRI findings of clear-cell and endometrioid ovarian carcinomas, as with TVUS, are often overlapped, characterized by large cystic heterogeneous mixed ovarian mass with mural nodules protruding into the cystic space. Nevertheless, morphological features such as a round mural nodule, a high height-to-width ratio, and a focal growth pattern may help distinguish the first from the second [79].

Further studies are vital to substantiate the appliance of MRI in EAOC diagnosis, most pertinently distinguishing between a benign endometrioma with atypical features and EAOC, particularly in the reproductive age.

### 6.5. Laparoscopy

Since most cases of EAOC are diagnosed at an early stage, there may be no laparoscopic difference with benign endometrioma. In such cases, diagnosis of malignancy is made by frozen section or postoperatively by pathology [28,29,31]. In those cases, the tumor is limited to one ovary, the capsule intact, without a tumor on the ovarian surface, no ascites, and no metastatic lesions. Surgical suspicion during laparoscopy should be raised when the ovary is significantly enlarged (above 9 cm) and the cystic tumor contains clear papillary or solid portions [30]. In more advanced EAOC cases, the ovary is interfused with solid parts with a papillary surface [30].

In this regard, bloody ascites and high serum CA-125 levels may be misleading during laparoscopy. In rare instances, advanced endometriosis may be associated with ascites, pleural effusions, and large pelvic masses, with no evidence of ovarian malignancy [80,81].

## 7. What Is the Prognosis of Women with EAOC?

In general, ovarian cancer is the second-most common gynecological malignancy in developed countries while having the highest mortality [43,82]. The overall 5-year survival for ovarian cancer is <50%, ranging from nearly 90% in stage IA to <20% in stage IV disease [83]. Despite advances in cytoreductive radical surgery and cytotoxic chemotherapy over the last few decades, only marginal improvement has resulted in these figures [43]. This has been attributed to the absence of effective early detection strategies and the rapid development of chemo-resistance.

Previous studies focusing on EAOC prognosis have resulted in contrasting results regarding the prediction of these women, primarily due to the small sample size of available publications [84]. The largest available study includes 159 women with patients self-reporting endometriosis diagnosis [85].

Recently, in a large, robust population-based Dutch nationwide study that combed two databases and assessed 32,419 patients with ovarian cancer, 1979 patients (6.1%) had histologically proven endometriosis [86]. The endometriosis group (apparently EAOC) cohort was younger at ovarian cancer diagnosis and had a more favorable grading and stage of disease (stage 1, 52.1 versus 18.5%). Furthermore, women in the endometriosis cohort more often had surgical treatment for ovarian cancer (97.8 versus 73.6%). They had more prolonged overall survival than women with ovarian cancer without endometriosis, even after adjusting for confounders. The median survival for the endometriosis cohort was 12.0 years (IQR, 3.0–not reached), and for the control cohort, 2.0 years (IQR, 1.0–8.5) (*p* < 0.0005). The crude hazard rate for overall survival was 0.46 (95% CI = 0.43–0.49), and after controlling for confounders (age, tumor grade, disease stage, and treatment), it was 0.89 (95% CI = 0.83–0.95) in favor of EAOC as compared to ovarian cancer with no endometriosis.

Similarly, in a recent systematic review and meta-analysis that included 21 eligible studies and 38,641 patients, EAOC had better overall and progression-free survival than controls [87].

Several explanations have been suggested to improve prognosis in women with EAOC. Women with endometriosis have physical symptoms that may urge them to seek earlier medical consultations, physical examinations, and targeted pelvic ultrasounds. In addition, women with endometriosis are frequently treated with continuous oral contraception, reducing the risk of ovarian cancer [88]. The more active immune system in women with endometriosis was also recently suggested as a possible factor for a better prognosis in EAOC patients [89]. Most captivatingly, EAOC is assumed to be a distinct entity of epithelial ovarian cancer with different pathophysiological and genetic backgrounds, as discussed in the pathogenesis section.

## 8. What Is the Clinical Management of an Intact Endometrioma in Reproductive Age?

An intact endometrioma in reproductive age may be challenging to manage in the clinical setting, especially in infertile women or women planning for a future pregnancy. Several vital issues should be examined and discussed individually for well-informed consent and proper management. Up to 50% of women with endometriosis may need ART treatment [90]. Surgical excision of an endometrioma is still a common and established treatment choice. However, the recurrence rate of endometriosis following surgery is high, estimated at 21.5% and 40–50% at two and five years, respectively [91]. Furthermore, the need for repeat operations seems high [92,93,94]. The risk of developing EAOC in these women may further complicate their surveillance and management.

From the reproductive endocrinologist’s perspective, several concerns are related to the quantitative and qualitative aspects of ovarian reserve and pregnancy attainment. Whether or not an endometrioma per se may harm the ovarian reserve continues to be controversial and debatable [95,96] and has been recently challenged [97]. Conversely, it is well-accepted nowadays that endometriotic cystectomy has a harmful and irreversible effect on ovarian reserve [97,98]. A recent systematic review and meta-analysis analyzed 12 prospective studies containing 783 women: 489 and 294 with unilateral and bilateral endometriomas, respectively [99]. Serum anti-Müllerian hormone (AMH) levels dropped significantly, by 1.65 ng/mL (95% CI = 1.15 to 2.15) and by 2.03 ng/mL (95% CI = 1.47 to 2.58), at 9–12 months postoperatively as compared to baseline in the unilateral and bilateral ovarian endometriotic cystectomy groups, respectively. This decline corresponds to a 39% and 57% decrease in the functional ovarian reserve following surgery, potentially suggesting a long-standing impact on these women’s reproductive life span. Furthermore, AMH was also shown to be a more sensitive and reliable ovarian reserve biomarker than antral follicle count (AFC) in women with intact endometrioma [99].

An intact endometrioma does not seem to disrupt ovulation or achieve natural pregnancy [100]. In addition, in the ART setting, an intact endometrioma does not seem to compromise clinical pregnancy and live-birth rates [101,102]. Furthermore, endometriotic cystectomy does not seem to improve clinical pregnancy and live birth rates in the ART setting [101,103].

More conservative and less invasive modalities of endometrioma treatment should be explored further, especially in infertile women or those planning for future pregnancies. Modalities that may ameliorate the impact on ovarian reserves, such as ultrasound-guided sclerotherapy or laser vaporization, should be pursued.

## 9. Conclusions

Endometriosis is a widespread disease in reproductive age, and endometrioma is a dominant manifestation, with estimated equivalents of up to 1:10 and 1:18 in women, respectively. Ovarian cancer developing at reproductive age raises much concern and trepidation, especially in women with infertility and others postponing live birth or planning future pregnancies. Furthermore, it challenges medical practitioners with practical issues related to the long-term management of these women through adulthood and past the menopausal transition.

EAOC is believed to develop merely from an endometrioma; however, further studies are essential to examine the link between superficial and deep infiltrating endometriosis. The main two entities of EAOC are epithelial, clear-cell ovarian cancer, and endometrioid cancer, consistent with 3.4- and 2.3-fold risk, respectively. The risk for low-grade serous tumors seems inconclusive.

According to 2017–2019 data from the National Cancer Institute, the lifetime ovarian cancer risk in the general female population is 1.1%. Considering the SRR of 1.93 in women with endometriosis, the calculated lifetime risk of EAOC is 2.1%. Accordingly, the lifetime risk of ovarian cancer in women with endometriosis increases from 1:91 to 1:48 women. To put it into perspective, these figures are interpreted as low compared to other lifetime cancer risks of breast, lung, and bowel: 1:8, 1:17, and 1:25, respectively. Since many methodological confounders and critical risks of bias are found in many previously published epidemiological data, further high-quality targeted studies are essential to substantiate the EAOC threat.

Generally, women with EAOC are older than women with benign endometrioma but younger than women with non-EAOC ovarian carcinomas EAOC, such as high-grade serous ovarian cancer. In the majority of cases, ovarian cancer is a postmenopausal disease. In about 12% of cases, ovarian cancer may develop in women < 44 years of age. This rate includes a substantial number of women with borderline and non-epithelial tumors.

The likelihood of EAOC during the reproductive period appears infrequent. About 50% of EAOC cases are diagnosed in premenopausal women, not necessarily during reproductive years. Notably, nearly half of women with EAOC do not have children upon diagnosis. More studies are essential to accurately calculate the real risk of EAOC at reproductive age.

The pathogenesis of the malignant transformation of endometriosis is still under active investigation. Genetic, inflammatory, immunologic, and hormonal factors have been implicated in this occurrence. The innovative dual paradigm of two distinct types of ovarian epithelial carcinomas with different molecular profiles, type I and type II, continues to prevail. According to this model, clear-cell and endometrioid carcinomas are sub-types of type I that have low-grade features and develop from endometriosis with distinct molecular profiles. Recently, state-of-the-art genetic methods reported a strong relationship between endometriosis and EAOC, suggesting causality. Additional targeted, high-quality studies employing advanced genetic methodologies are imperative to substantiate the causal relationship between endometriosis and EAOC.

The clinical diagnosis of EAOC at reproductive age is challenging. Endometrioma is a frequent finding in women with endometriosis, while EAOC is a rare occurrence. Clinical symptoms and signs may be similar in both entities, and routine employment of ovarian cancer or other genetic biomarkers has yet to be further explored. TVUS is the best imaging modality to differentiate between benign endometrioma and EAOC. The expertise of the TUVS performer is of high significance in atypical cases. Advanced age, nulliparity, hyperestrogenism, and large-sized endometrioma above 9 cm are prominent features that increase the risk of EAOC. In cases of atypical endometrioma features on TVUS (estimated at 5–25% of cases), MRI has an essential supportive role. In most of these cases, the tumors are usually benign, and MRI may be necessary to reduce patient anxiety, elude repeat imaging, and avoid surgery.

Women with EAOC are diagnosed at an earlier stage and have a more favorable histological grade than other ovarian forms of ovarian cancer with no endometriosis. Furthermore, they have better progression-free and overall survival rates. These estimates are validated following adjustment for confounders.

Surgical treatment of benign endometrioma, specifically endometriotic cystectomy, continues to be an acceptable approach to therapy in reproductive age. However, a more conservative approach is starting to permeate among practitioners, especially in infertile women, and others postponing live birth or planning for a future pregnancy, mainly due to the adverse surgical impact on ovarian reserve and reproductive life span potential.

In women postponing live birth or planning for a future pregnancy with an endometrioma, it is mandatory to clarify the rationale for early parenthood. Otherwise, prolonged oral contraception or progestin therapy may reduce EAOC risk. Infertile women with intact endometrioma, a manifestation of advanced disease, should be counseled for ART.

In these women, particularly in nulliparous, the appearance of an atypical endometrioma on TVUS suggesting an EAOC, especially a large endometrioma with a solid component, requires MRI performance. Since each case has its own specific characteristics, a multi-disciplinary discussion and appropriate patient counseling should be conducted to reach an optimal therapeutic plan. Surgery and histologic evaluation may be inevitable for final diagnosis and treatment in these cases.

Since most EAOC cases are diagnosed early with a favorable prognosis, optimal cytoreductive with fertility-sparing surgery may be feasible. The pros and cons of fertility preservation techniques in these women, including oocyte or ovarian tissue cryopreservation, should be discussed.

## Figures and Tables

**Table 1 biomolecules-12-01721-t001:** Potential confounding factors that may impact risk estimation of endometriosis-associated ovarian carcinoma.

Confounding Factor	Clarification	Reference	Methodology and Sample Size	Conclusion
Type of cancer	Gonadal versus extra-gonadal	Kavaskoff et al., 2021 [16]	Systematic review and meta-analysis including 49 population-based case-control and cohort studsies	Summary of Relative Risk
				EAOC 1.99
				Thyroid cancer 1.39
				Breast cancer 1.04
				Colorectal cancer 1.0 (NS)
				Endometrial cancer 1.29 (NS)
				Cutaneous melanoma 1.17 (NS)
				Cervical cancer 0.68
Histological type	Serous, mucinous, endometrioid, or clear-cell ovarian cancer	Kavaskoff et al., 2021 [16]	Systematic review and meta-analysis including 49 population-based case-control and cohort studies	Summary of Relative Risk
				Clear-cell 3.44
				Endometrioid 2.33
				Serous 1.17
				Mucinous 0.98 (NS)
				Borderline 1.46
Endometriosis sub-type	Superficial, endometriotic cyst, or deep infiltrating disease	Saavalainen et al., 2018 [17]	A population-based study including 49,933 surgically verified endometriosis, comprising ovarian (*n* = 23,210), peritoneal (*n* = 20,187), and deep infiltrating (*n* = 2372)	Standardized Incidence Ratio of Ovarian Cancer
				Endometrioma: 2.56
				Peritoneal: 1.32 (NS)
				Deep infiltrating 1.41 (NS)
Different countries and geographic regions	Clear-cell ovarian carcinoma is more prevalent in Southeast Asia	Machida et al., 2019 [18]	A nationwide retrospective (comparative) registry study performed between 2002 and 2015. Japan cohort 48,640 and USA cohort 49,936 cases.	Histological Subtype Incidence in Japan versus USA
				Clear-cell: 26.9% versus 8.4%
				Endometrioid: 19.2% versus 14.8%
Targeted cohort	Infertility is associated with a higher risk of cancer	Murugappan et al., 2019 [20]	Retrospective cohort analysis between 2003 and 2016 using an insurance claims database, including 64,345 infertile compared to 3,128,345 non-infertile women.	Adjusted Hazard Ratio in Infertile Compared to Fertile Women
				Overall cancer: 1.18
				Ovarian cancer: 1.64
Endometriosis diagnosis	Surgical (histological) versus imaging versus self-reported	Shafrir et al., 2021 [21]	Comparison between questionnaire-reported endometriosis with medical record notation among participants from five international cohorts. The baseline population included 405,898 women. The total number of eligible women who self-reported endometriosis was 5131.	Confirmation was 84% overall when combining clinical, surgical, and pathology records, ranging between 72–95% among the assessed cohorts.
Previous endometriosis treatment	Medical or surgical	Iversen et al., 2018 [21]	Prospective, nationwide cohort study from 1995 to 2014. All women 15–49 years of age were eligible. Final study population included 1,879,227 women.	Reduced Risks of Ovarian Cancer Compared with Never-Users
				Current users: 0.58
				Recent users: 0.77
				Little evidence of major differences in risk estimates by tumor type or progestogen; content of combined oral contraceptives was seen.
				Use of hormonal contraception prevented 21% of ovarian cancers in the study population.
Methodological risk of bias	In systematic reviews and meta-analyses	Kvaskoff et al., 2021 [16]	Systematic review and meta-analysis, including 49 population-based case-control and cohort studies	26 studies were scored as having a ‘serious’/‘critical’ risk of bias, and the remaining 23 as ‘low’/‘moderate’.

EAOC—endometriosis-associated ovarian cancer; NS—not significant.

**Table 2 biomolecules-12-01721-t002:** Ultrasonographic classification systems employed in the clinical setting to categorize the risk of malignancy.

A: The International Ovarian Tumor Analysis Simple Rules [67].
Benign features:
A unilocular cyst (any size)
No solid components, or solid components < 7 mm in diameter
Presence of acoustic shadowing
Smooth multilocular cyst < 10 cm in diameter
No blood flow
Malignant features:
Irregular solid tumor
Ascites
≥4 papillary structures
Irregular solid multilocular tumor, with the largest diameter >10 cm
Very strong-colored Doppler flow
**B: The American College of Radiologists O-RADS System for classification of adnexal lesions. Category description and risk of cancer [68].**
O-RADS 1 Normal ovary (no risk of cancer)
O-RADS 2 Almost certainly benign lesion (< 1% chance of cancer)
O-RADS 3 Low-risk lesion (1 to < 10% chance of cancer)
O-RADS 4 Intermediate-risk lesion (10 to 50% chance of cancer)
O-RADS 5 High-risk lesion (>50% chance of cancer)

## Data Availability

No new data were generated in support of this manuscript.

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
