# Peer review of "Endometriosis-Associated Ovarian Cancer: What Are the Implications for Women with Intact Endometrioma Planning for a Future Pregnancy? A Reproductive Clinical Outlook"

_biomolecules, 2022, doi:10.3390/biom12111721_

Round 1
Reviewer 1 Report
I understand that this article has been submitted to a special issue of the Journal Biomolecules entitled “Molecular and Cell Biology in Endometriosis and Endometrial Cancer”. At first sight, as indicated in “Materials and methods”, the manuscript seems more of a clinical and epidemiological nature than a review dealing with structures and functions of bioactive and biogenic substances, or on molecular mechanisms. In this connection, the site of the Journal clearly states that it accepts articles “on structures and functions of bioactive and biogenic substances, molecular mechanisms with biological and medical implications as well as biomaterials and their applications”. Clearly, section 5 deals with pathogenetic mechanisms, but the section is meant to better understand the clinical implications, not vice versa.
If this article is to be published in Biomolecules in its entirety, then the most important section is that on pathogenesis. Therefore, it is this section that should be expanded. This is because, unless a clear cause-effect relationship can be established, the association remains speculative.
In terms of clinical management, section 8 deals i.a., with laparoscopic excision. I believe that mention of differential features observable at laparoscopy between an endometrioma and EAOC would be helpful.
Lines 43-44: The statement that “Endometrioma represents a more advanced stage of the disease” needs to be further specified. Without going back to the early debate between Sampson [Sampson JA. The life history of ovarian hematomas (hemorrhagic cysts) of endometrial (Müllerian) type. Arch Surg. 1922;5:217-80] and Bailey [Bailey KV. The etiology, classification and life history of tumours of the ovary and other female pelvic organs containing aberrant Müllerian elements, with suggested nomenclature. J Obstet Gynaecol Br Emp. 1924;31:539-79], on whether the ovary becomes affected primarily or secondarily, it is unclear what the sentence means. Does the Author imply that 55% of women with superficial peritoneal endometriosis eventually develop a monolateral ovarian endometrioma and 28% endometriomas bilaterally?
Lines 54-63: This statement needs one or more references, or a mention that the issue will be discussed in section 3 below. Indeed, The Author puts the risk of malignancy in women with endometriosis in its proper perspective in lines 108-115
Reviewer 2 Report
biomolecules-1963715
This is a narrative review of endometriosis-associated ovarian cancer (EAOC). Some aspects can be improved to make it more comprehensive.
1. More information and details of those studies should be added in Table1, such as sample size and conclusion.
2. Some references are not relevant to this topic. For example:
2-1. Ref 16: It reported 46 patients with malignant transformation of abdominal wall endometriosis (AWE), but seems not contained a risk evaluation.
2-2. Ref 17: not a risk estimation paper.
2-3. Ref 59: this paper only tells “Hypermethylation of the RUNX3 was related to the malignant transformation of endometriosis and that this process was related to corresponding changes in the eutopic endometrium. Furthermore, the ‘oestrogen-DNMT1’ signaling pathway may induce the hypermethylation of RUNX3 to promote the malignant transformation of endometriosis”, not “Unopposed estrogen stimulation, without the modulating protection of progesterone, seems to be one of the highest risk factors associated with the malignant transformation of endometriosis “(lines 247-249).
